# Design and Analysis of Systematic Batched Network Codes

**DOI:** 10.3390/e25071055

**Published:** 2023-07-13

**Authors:** Licheng Mao, Shenghao Yang, Xuan Huang, Yanyan Dong

**Affiliations:** 1School of Science and Engineering, The Chinese University of Hong Kong, Shenzhen, Shenzhen 518172, China; lichengmao@link.cuhk.edu.cn; 2Department of Information Engineering, The Chinese University of Hong Kong, Hong Kong, China; 1155136647@link.cuhk.edu.hk; 3Department of Electrical and Computer Engineering, National University of Singapore, Singapore 117597, Singapore

**Keywords:** network coding, systematic code, random linear network coding, batched network coding, BATS code

## Abstract

Systematic codes are of important practical interest for communications. Network coding, however, seems to conflict with systematic codes: although the source node can transmit message packets, network coding at the intermediate network nodes may significantly reduce the number of message packets received by the destination node. Is it possible to obtain the benefit of network coding while preserving some properties of the systematic codes? In this paper, we study the systematic design of batched network coding, which is a general network coding framework that includes random linear network coding as a special case. A batched network code has an outer code and an inner code, where the latter is formed by linear network coding. A systematic batched network code must take both the outer code and the inner code into consideration. Based on the outer code of a BATS code, which is a matrix-generalized fountain code, we propose a general systematic outer code construction that achieves a low encoding/decoding computation cost. To further reduce the number of random trials required to search a code with a close-to-optimal coding overhead, a triangular embedding approach is proposed for the construction of the systematic batches. We introduce new inner codes that provide protection for the systematic batches during transmission and show that it is possible to significantly increase the expected number of message packets in a received batch at the destination node, without harm to the expected rank of the batch transfer matrix generated by network coding.

## 1. Introduction

Network coding has great advantages compared with the traditional store-and-forward in network communications [1,2,3]. Random linear network coding (RLNC) provides a decentralized approach to network coding and achieves the multicast capacity of networks with packet loss in a broad setting [4,5,6,7,8,9,10]. In the past twenty years, extensive studies have been performed towards resolving the implementation issues of RLNC, such as the computational complexity and the coefficient overhead [11,12,13,14]. Batched network coding extends RLNC by introducing an inner code–outer code structure [15,16,17,18,19,20,21]. In particular, the outer code of a batched network code encodes the message packets into a sequence of batches, each of which is a number of coded packets, and the inner code is formed by linear network coding applied on the coded packets belonging to the same batch. The design of the outer code and the inner code can be separated, where the outer code achieves end-to-end reliability and the inner code maximizes the network efficiency [22]. The number of packets in a batch (called the *batch size*) affects the coefficient overhead and the computational complexity. To achieve the benefits of network coding and constrain the overhead/complexity, the batch size is usually a small integer larger than 1, e.g., 8 or 16 [23]. Batched network coding allows joint batch encoding/decoding, while the original RLNC schemes can be regarded as special batched network codes where the outer code has a single batch or multiple batches encoded/decoded separately.

In coding theory, a code is said to be systematic if all message symbols form a subset of the coded symbols [24]. Many practical codes can be designed to be systematic—for example, Reed–Solomon codes [25], fountain codes [26] and polar codes [27]. Standardized LDPC codes in both 802.11 and 5 G NR are systematic. For network communications, the retransmission-based end-to-end reliability scheme can be regarded as a systematic code. The benefits of the systematic codes are also attractive for practical applications of batched network codes, especially for latency-sensitive applications [28,29,30].

Different from systematic channel coding, systematic batched network coding needs to take both the outer code and the inner code into consideration. Though not optimal in general, an overlapping outer code with batches formed by subsets of the message packets proposed in [15,16,17,20] is already systematic in the sense that the union of some batches includes all the message packets. However, even with a systematic outer code, the benefits of systematic codes cannot be obtained due to network coding: using random linear coding at the intermediate nodes prevents the reception of the message packets at the destination nodes. The problem cannot be solved by simply excluding the message packets from network coding, which reduces the benefits of network coding. Most existing works on systematic RLNC focus on encoding and decoding at the source node and destination nodes, respectively, without considering network coding at the intermediate nodes [31,32,33,34,35,36].

In this paper, we study systematic batched network codes that have a systematic outer code and an inner code that can preserve the benefits of the systematic outer code. Our contributions are summarized as follows.

### 1.1. Contributions Regarding Systematic Outer Codes

The outer code of a batched network code can be designed by extending fountain codes or LDPC codes [19,21], which achieve higher rates than the overlapping outer codes for the same inner code. The existing outer codes obtained by coding are not designed to be systematic. In principle, any linear code can be systematic by transforming the generator matrix to the reduced echelon form. As the existing outer codes obtained by coding are linear, they can also be systematic. The main issue, however, is how to preserve the low encoding/decoding computation cost: a general transformation of the generator matrix by Gaussian elimination affects the structure of the codes and hence may increase the computation cost.

In this paper, we design a systematic outer code based on the BATched Sparse (BATS) outer code, which is a matrix-generalized fountain code [19]. When the batch size is 1, the BATS outer code becomes a fountain code. The BATS outer code preserves the rateless feature of fountain codes, i.e., the number of batches that can be generated is unlimited (i.e., the rateless property) and can achieve a nearly optimal outer code rate with low encoding/decoding complexity. To preserve the salient features of the BATS outer code, the systematic outer code is also expected to be rateless, where the first $n_{s}$ batches (called *systematic batches*) consist of a partition of the message packets. In addition to the systematic batches, the outer code can further generate more batches, called *non-systematic batches*. The fountain code has a low-complexity systematic design that benefits from the universal degree distribution [37]. When the batch size is larger than 1, the degree distribution of the BATS outer code depends on the rank distribution of the batch transfer matrices and hence is not universal. For this reason, the systematic design of BATS codes has to consider some new issues that do not appear in the systematic fountain code design.

In this paper, we generalize the fountain code approach to design a systematic outer code, which uses a (non-systematic) BATS outer code that satisfies the *consistency* requirement. In particular, a consistent outer code generates the first $n_{s}$ batches deterministically, which can recover all the message packets. To ensure a small coding overhead, $n_{s}$ is expected to be as small as possible. For a fountain code, the minimum value of $n_{s}$ is the same as the number of message packets, and a consistent code with the minimum value of $n_{s}$ can be found using a number of trials of the random encoding procedure of the fountain code. As fountain codes are universal, for each number of message packets, a consistent code can be designed once and used forever. However, BATS outer codes are not universal, and, even for the same number of message packets, the consistent code is different for different rank distributions. Our experiments show that when the number of message packets is larger, many more random trials are required to find a consistent outer code with a small coding overhead.

To design a systematic outer code with a small value of $n_{s}$ more efficiently, we propose a structured encoding approach for the first $n_{s}$ batches, called *triangular embedding*. Using triangular embedding, zero-coding-overhead outer codes can be designed with one or two random trials for a large range of the number of message packets. Triangular embedding does not increase the computation costs of both encoding and decoding. Moreover, we also verify in experiments that the batches generated by triangular embedding can be used with the batches generated by the BATS outer code and demonstrate superior decoding performance compared to the BATS outer code.

We also analyze the encoding and decoding computation costs of the proposed systematic outer code. For encoding, the systematic outer code has a lower computation cost than the corresponding BATS outer code. The decoding computation cost of the systematic outer code depends on the number of message packets received at the destination node. When all the message packets are received, no computation is required for decoding. When some of the message packets are not received, the decoding computation cost of the systematic outer code increases with the number of message packets that are not received and is at most 2 times the computation cost of the BATS outer code decoding.

### 1.2. Contributions Regarding Inner Codes

We further study the inner code that can protect the message packets in the systematic batches. For line networks, *systematic inner coding* has been discussed for batched network coding [23], where an intermediate node transmits both the received packets and the recoded packets generated by linear combinations of the received packets. For a line network without packet loss, the destination node can receive all the message packets generated by the systematic outer code when using systematic recoding. However, if the packet loss rate for each communication link is bounded below by a positive number, the number of message packets that can be received by the destination node decreases exponentially rapidly as the network length increases. For systematic RLNC, a decode–recode network coding approach has been proposed to protect the message packets [38], where an intermediate node first tries to decode the message packets and then transmits the decoded message packets together with some recoded packets. Systematic RLNC is a special systematic batched network code with only the systematic batches, and the decode–recode approach is mainly discussed for extended window recoding.

In this paper, we extend and refine the decode–recode approach for the inner code of batched network coding. For a general batched network code, it is not necessary that the received packets of a batch at an intermediate node can decode all the original packets. In other words, the batch transfer matrix formed by the coefficient vectors of all the received packets of a batch at an intermediate node may have a rank lower than the batch size. We instead study how to decode some of the message packets uniquely at an intermediate node. We say that a message packet in a systematic batch is *recoverable* at an intermediate node if it can be uniquely solved by the received packets of the batch at the intermediate node. We give a necessary and sufficient condition such that a message packet in a batch is recoverable, and we show that using Gauss–Jordan elimination, we can find all the recoverable message packets in a batch. We also analyze the recovery of the message packets at the next hop subject to packet loss and side information. Our analysis shows that generating all recoded packets using random linear coding is not preferable, and knowing more information about recoding than the coefficient vectors does not aid in the recovery of message packets.

Based on our analysis, we improve systematic inner coding to protect the message packets in a batch, where the level of protection can be tuned by a parameter. Our inner codes can achieve the same network coding gain as the existing inner codes, while significantly improving the number of received message packets. By tuning the parameter, the number of received message packets can be further increased with the cost of lower coding rates. Both the recovery of the message packets and the message protection recoding are linear operations on a batch, and hence our inner code does not increase the coefficient overhead for decoding at the destination node.

### 1.3. Paper Organization

The remainder of this paper is organized as follows. Section 2 is a self-contained introduction of batched network coding with the BATS outer code. In Section 3, we propose a general approach to systematic outer codes based on the BATS outer code. In Section 4, we introduce the triangular embedding approach to improve the design efficiency of the systematic outer code. In Section 5, we discuss the inner coding schemes that can protect the message packets in systematic batches. Section 6 presents the concluding remarks.

## 2. Ordinary Batched Network Coding

We briefly introduce ordinary (non-systematic) batched coding to assist the further discussion of the systematic design. A batched network code is formed by an outer code and an inner code. Here, we focus on a specific outer code called the BATS outer code, which was originally introduced by the BATS code. Readers are referred to [23] for more information about the BATS code.

### 2.1. BATS Outer Code

The outer code introduced here is also called the *ordinary* outer code, in contrast to the systematic outer code, to be discussed in the next section.

A finite field of size *q*, denoted as $F_{q}$, is called the base field. A packet of length *T* is a column vector in $F_{q}^{T}$, and a set of packets of the same length is equated to the matrix formed by juxtaposing the packets in the set. We consider the transmission of *K* message packets, which form the T×K matrix B from the source node to the destination node in a network.

The (ordinary) outer code encodes the *K* message packets in two steps. The first step uses a systematic precode to generate a number of redundant packets, which are also called *parity check packets*. Let $K^{\prime }\geq K$ be the total number of packets containing the message packets and the parity check packets. Denote by $B_{p}$ the $K^{\prime }-K$ parity check packets. Let P the $K^{\prime }\times \left(K^{\prime }-K\right)$ parity check matrix of the precode, i.e.,
(1)$$[B\;\;B_{p}]P=0.$$

The parity check packets can include both low-density parity check (LDPC) and high-density parity check (HDPC) packets to balance the computation cost and the decoding performance. Refer to [37] for such a design of P.

Let $B^{\prime }=\left[B\;\;B_{p}\right]$, which are called the precoded packets. The second encoding step of the outer code generates batches of coded packets. Let *M* be a positive integer called the *batch size*, which is usually less than a hundred. For i=1,2,…, the *i*th batch $X_{i}$ includes *M* packets generated from a subset $B_{i}\subset B^{\prime }$ as follows:$$X_{i}=B_{i}G_{i},$$
where $G_{i}$ is a matrix of *M* columns called the *batch generator matrix*. The number of packets in $B_{i}$, which is also the number of rows of $G_{i}$, will be specified later. When M=1, the outer code becomes a fountain code. The design of $B_{i}$ is discussed as follows.

Here, we discuss general batch encoding that can be used for various decoding approaches, including inactivation decoding. The precoded packets are further separated into two parts:*active packets* that include a subset of the message packets and all the LDPC packets, and*inactive packets* that include all the other message packets and all the HDPC packets.

Denote by *A* the number of active packets. Then, the number of inactive packets is $K^{\prime }-A$. We require A≥K. As a special case, when there are no HDPC packets or inactive packets during encoding, we have $A=K^{\prime }$. The encoding of a batch uses both active and inactive packets.

The number of active packets used in a batch is determined using a degree distribution $\Psi =(\Psi_{1},\ldots ,\Psi_{D_{\max}})$, and it affects the decoding performance of both belief propagation decoding and inactivation decoding. The degree distribution Ψ is designed based on the *batched transfer matrix rank distribution* induced by the inner code. The maximum number $D_{\max}$ for the active packets is sufficient to be a couple of multiples of *M*, as proven in [19]. For the encoding of each batch $X_{i}$,

Independently sample Ψ and obtain an integer $d_{i}^{A}$, which is called the active degree of the batch;Uniformly, at random, choose $d_{i}^{A}$ active packets to be included in $B_{i}$.

The inactive packets can help to further improve the inactivation decoding performance. When M=1, on average, each batch may involve 2 or 3 inactive packets [26]. When M>1, the number of inactive packets in a batch can be $3(K^{\prime }-A)/n$, where *n* is the number of batches expected to be used for decoding. Denote by $d_{i}^{B}$ the number of inactive packets used in the *i*th batch.

Considering both active and inactive packets, $B_{i}$ has $d_{i}=d_{i}^{A}+d_{i}^{B}$ packets, where $d_{i}$ is called the total degree of the batch. $G_{i}$ is a $d_{i}\times M$ uniformly random matrix with entries from the base field. In practice, random encoding can be implemented by a pseudorandom number generator. The random values in the encoding process can be used for decoding if they share the same pseudorandom number generator at the source node and the destination node.

Denote by ENC the encoder that implements the above encoding process of the BATS outer code. The pseudocodes of ENC are given in Appendix C for reference.

### 2.2. General Inner Code Formulation

We use a line network as an example to introduce the inner code, and the inner code can be extended to other network typologies as discussed in [23]. A line network of length *L* is formed by a sequence of network nodes labeled by 0,1,…,L, where the first node 0 is the source node and the last node *L* is the destination node. All the other nodes are called intermediate nodes. Network links exist only between two consecutive network nodes, modeled by packet erasure channels, i.e., a packet transmitted on a network link is either correctly received or erased. Figure 1 illustrates the line network.

The inner code is the composition of the *recoding* operations performed on each batch separately. The recoding at the source node takes the batches generated by the outer code as the input, and the recoding at an intermediate node takes the received packets of a batch as the input. For each batch, recoding generates a number of linear combinations of the packets belonging to the batch, and the packets generated by recoding are supposed to belong to the same batch. There are various approaches to the recoding operation, which is determined by the linear combination coefficients. The original RLNC schemes use coefficients chosen uniformly at random from the base field [4,6,7], and extensive research has been carried out towards recoding with lower complexity and latency [39,40,41,42,43,44]. In this paper, we study the recoding schemes that can fulfil the systematic coding requirement.

Without specifying a recoding scheme, we give a general formulation of recoding. Fix a certain network node *u*. Let $Y_{i}^{\left(u\right)}$ be the received packets of the *i*th batch at the node *u*. At the source node, $Y_{i}^{\left(0\right)}=X_{i}$. As recoding is linear, for v=1,…,L,
(2)$$Y_{i}^{\left(u\right)}=X_{i}H_{i}^{\left(u\right)}=B_{i}G_{i}H_{i}^{\left(u\right)},$$
where $H_{i}^{\left(u\right)}$ is called the *(batch) transfer matrix* of the *i*th batch at the node *u*. The number of rows of $H_{i}^{\left(u\right)}$ is *M*. The number of columns of $H_{i}^{\left(u\right)}$ corresponds to the number of packets received for the *i*th batch at the node *u*, which may vary for different batches and is finite. If no packets are received for a batch, $Y_{i}^{\left(u\right)}$ ($H_{i}^{\left(u\right)}$) is the empty matrix of 0 columns.

Note that the transfer matrices are determined not only by the recoding scheme, but also by the network packet loss pattern. Due to the randomness in both recoding and packet loss, the transfer matrices cannot be derived from the recoding design. To obtain the transfer matrices, RLNC introduces *coefficient vectors* embedded in the packet header immediately after $X_{i}$ is generated. The matrix formed by the coefficient vectors is the identity matrix. The same linear operations performed on a batch are performed on the coefficient vectors as well, so that $H_{i}^{\left(u\right)}$ can be known at each node *u* that receives batch *i* from the header of the batch.

We say that a set of packets of a batch are linearly independent/dependent if their corresponding coefficient vectors in the packet header are linearly independent/dependent. We call $\mathrm{rank}(H_{i}^{\left(u\right)})$ the *rank of the ith batch* at node *u*.

### 2.3. Decoding Algorithms

Suppose that *n* batches $Y_{i}^{\left(L\right)}$, i=1,…,n are received at the destination node *L*. A decoder is expected to recover B using $Y_{i}^{\left(L\right)}$, i=1,…,n, which are related by a linear system. From this perspective, we obtain an upper bound on the decoding performance [23]:$$K\leq \sum_{i=1}^{n}\mathrm{rank}\left(H_{i}^{\left(L\right)}\right).$$

When used as a block code with a fixed number *n* of batches, the *(outer) coding rate* defined as K/n, together with the decoding success probability, is used to measure the outer code performance. When used as a rateless code, decoding allows more batches to be used until all the message packets are decoded, and the *(outer) coding overhead* defined as $\sum_{i=1}^{n}\mathrm{rank}\left(H_{i}^{\left(L\right)}\right)-K$ is used to measure the decoding performance.

As B and $Y_{i}^{\left(L\right)}$, i=1,…,n are related by a linear system, Gaussian elimination is the optimal algorithm to solve B. However, Gaussian elimination incurs a computational complexity linear in *K* when decoding one message packet on average, which is not tolerable when *K* is slightly large. In the remainder of this section, we introduce several approaches that can achieve O(1) complexity in decoding one message packet. In the following, we first discuss two decoding algorithms without inactive packets and then discuss inactivation decoding.

#### 2.3.1. Two-Step Decoding

Suppose that the number of inactive packets during encoding is 0, so that $d_{i}^{B}=0$ for all batches. We first discuss the two-step decoding approach. The first step recovers a fraction $\eta \geq K/K^{\prime }$ of precoded packets using a belief propagation (BP) algorithm, which repeats the following operations:A batch *i* is said to be decodable if $d_{i}^{A}=\mathrm{rank}\left(G_{i}H_{i}^{\left(L\right)}\right)$; solve a decodable batch by Gaussian elimination;Substitute the decoded (precoded) packets into other undecoded batches and update the corresponding batch degree and generator matrix.

The BP decoding algorithm has a low computation cost that does not depend on the total number of message packets *K*. The second step decodes the precoded packets to recover the message packets, which is expected to be successful if the first step recovers at least η fractions of all the precoded packets.

Assume that the ranks of batch transfer matrices at the destination node $\mathrm{rank}(H_{i}^{\left(L\right)})$ are i.i.d and follow the distribution $h=(h_{0},h_{1},\ldots ,h_{M})$. We call $E\left[h\right]=\sum_{i=1}^{M}ih_{i}$ the expected rank. According to the theory of BATS codes [23], it is possible to design a degree distribution Ψ for a given rank distribution h such that when *K* is large, the BP decoding can recover a given η fraction of the precoded packets with a high probability when the coding rate K/n is larger, but very close to E[h]. In other words, we only need slightly more than K/E[h] batches to recover the *K* message packets.

#### 2.3.2. Joint Decoding

The above two-step decoding algorithm can be improved by combining the two steps when the precoding includes LDPC. For LDPC precoding, each parity check constraint can be regarded as a batch with batch size 1 and only one all-zero received packet. Then, the BP decoding of the batches in the first step of the two-step approach can also include the parity checks.

In practice, the decoding of the LDPC precode and the decoding of the batches in the two-step decoding algorithm can be combined together to improve the performance. The joint decoding algorithm can improve the decoding success rate and reduce the coding overhead of the two-step decoding algorithm, but does not increase the computation cost of the two-step decoding.

#### 2.3.3. Inactivation Decoding

When *K* is relatively small or the coding overhead is small, BP decoding tends to stop before decoding all the message packets. Although we can continue decoding by Gaussian elimination, the computational complexity is high.

A better approach is to use *inactivation decoding*: when BP decoding stops, an undecoded message packet is marked as inactive and substituted into the batches as a decoded packet to resume the BP decoding procedure. The decoding of batches with inactive packets also induces linear constraints on the inactive packets. Eventually, all the message packets are either decoded or inactive. The inactive packets are then solved by the linear constraints induced by decoding batches and the precodes. Inactivation decoding has the same decoding performance as Gaussian elimination, but can have a much lower computation cost if the number of inactive packets is small.

Moreover, when using inactivation decoding, we can use the inactive packets during encoding. Inactive packets during encoding are treated as inactive from the beginning of inactivation decoding and hence are also called *pre-inactive packets*. The extra inactive packets added during decoding are called the *dynamic inactive packets*. See [23] for a detailed discussion of inactivation decoding for BATS codes.

Denote by DEC the decoder that implements one of the above decoding processes of the BATS outer code. The pseudocodes of DEC for two-step decoding are given in Appendix C for reference.

## 3. Systematic Outer Codes

In this section, we design a systematic outer code that can preserve the silent features of the ordinary BATS outer code. We call those batches that are designed to include all the message packets the *systematic batches*.

### 3.1. Naive Approaches

Before introducing our approach, we first discuss some naive approaches and their limitations. For a fixed number *n* of batches, the outer code is a linear block code and hence the encoding process can be described as
$$\left[\begin{array}{ccc} X_{1} & \cdots & X_{n} \end{array}\right]=B\tilde{G}$$
where $\tilde{G}$ is the K×nM generator matrix of the first *n* batches. Suppose that nM≥K. If $\tilde{G}$ has *K* columns forming the identity matrix, the outer code is systematic.

First, we show that the random encoding of the ordinary BATS outer code is not a systematic code with high probability. For a batch of total degree *d*, the probability that a coded packet is equal to a precoded packet is $dq^{-d}$. As not all precoded packets are message packets, the probability that a coded packet is equal to a message packet is no greater than $dq^{-d}$. Typically, d≥M≥2 and q=256. Thus, it is unlikely that a message packet appears in a batch using the ordinary outer code.

When *n* is slightly larger than K/M, the matrix $\tilde{G}$ obtained from the ordinary BATS outer code has rank *K* with a high probability. The general procedure to make a linear code systematic is to transform $\tilde{G}$ by elementary row operations into the reduced row echelon form. Although a systematic code can be obtained, the drawback of this approach is that the low encoding/decoding computation cost of the BATS outer code cannot be preserved.

Now, we discuss another naive approach that seems solve the computation cost issue. To simplify the discussion, suppose that the number of message packets *K* is a multiple of the batch size *M*. In this naive approach, the first K/M batches form a partition of all the message packets, and more (non-systematic) batches are generated according to the encoding of batches as an ordinary outer code discussed in Section 2.1. However, to guarantee good decoding performance using the naive approach, a high degree must be applied to all the non-systematic batches.

We show two cases wherein a high degree of the non-systematic batches is necessary. In the first case, one systematic batch is completely erased during the communication and all the other systematic batches are received by the destination nodes, together with a non-systematic batch. Suppose that the erased batch is randomly chosen. For all the received batches, the batch transfer matrix is the M×M identity matrix so that the decoding problem becomes one of traditional erasure coding. The total number of received packets is *K*. To guarantee the decoding of all the message packets, it is necessary that the degree of the received non-systematic batch is *K*.

In the second case, we consider that for *M* systematic batches, only one packet is erased during communication and all the other packets are received correctly. In other words, the batch transfer matrix of these *M* systematic batches is the M×M identity matrix with one column removed, chosen uniformly at random. The destination node also receives all the other systematic batches, together with a non-systematic batch, all with the identity batch transfer matrix. The total number of received packets is *K*. To guarantee the decoding of all the message packets, it is necessary that the degree of the received non-systematic batch is *K*.

From these cases, we see that to achieve a high coding rate using the naive approach, the degree of the non-systematic code must be high and hence the encoding/decoding complexity is high. In the remainder of this section, we derive an approach to obtain a systematic outer code that has similar encoding/decoding complexity to the ordinary BATS outer code.

### 3.2. General Approach to Systematic Outer Codes

We give a general approach tp systematic outer codes, which extends the idea of systematic fountain codes [37]. Suppose that we have *K* message packets B for encoding using a systematic outer code with batch size *M*, where *K* is not necessarily a multiple of *M*. Let $n_{s}$ be an integer larger than or equal to K/M, to be decided later. We wish to design an outer code such that the first $n_{s}$ batches are systematic batches that include all the message packets.

Our approach to a systematic outer code uses an ordinary outer code (ENC,DEC), where ENC is the encoder and DEC is the decoder, as described in Section 2.1 and Section 2.3, respectively. The encoder ENC has two parts ${\mathrm{ENC}}_{n_{s}}$ and ${\mathrm{ENC}}_{n_{s}^{+}}$, where ${\mathrm{ENC}}_{n_{s}}$ generates the first $n_{s}$ batches and ${\mathrm{ENC}}_{n_{s}^{+}}$ generates all the further batches. The decoder DEC in general applies to all the batches subject to any batch transfer matrices. We denote by ${\mathrm{DEC}}_{n_{s}}$ the case of DEC when applying to the first $n_{s}$ batches with the rank-*M* batch transfer matrices.

To construct the systematic outer code, we require (ENC,DEC) satisfying some additional requirements. The pair (ENC,DEC) is said to be *consistent* if the following conditions are satisfied:${\mathrm{ENC}}_{n_{s}}$ and ${\mathrm{DEC}}_{n_{s}}$ are deterministic; andfor any *K* packets B,
(3)$$B={\mathrm{DEC}}_{n_{s}}\left({\mathrm{ENC}}_{n_{s}}\left(B\right)\right).$$

For the consistency requirement 2, it is possible to verify (Equation 3) without any specific value B of *K* packets, i.e., it is not necessary to check all choices of *K* packets. The reason is that both ${\mathrm{ENC}}_{n_{s}}$ and ${\mathrm{DEC}}_{n_{s}}$ are linear operations and, if the decoding is successful, their joint effect is to multiply the K×K identity matrix. We discuss how to design a consistent outer code later. Here, we focus on how to use it to construct a systematic outer code.

For a consistent (ENC,DEC), the decoder ${\mathrm{DEC}}_{n_{s}}$ solves *K* message packets from the $n_{s}M$ coded packets generated by ${\mathrm{ENC}}_{n_{s}}$. Among the $n_{s}M$ coded packets, $n_{s}M-K$ coded packets are redundant and can be removed without affecting the decoding performance (The decoding of a BATS code requires us to solve a system of linear equations by elementary equation operations. Each coded packet corresponds to an equation of the system. Each equation can solve at most one message packet. Therefore, exactly *K* equations are eventually transformed into the solutions of the message packets. The other equations are redundant). All the redundant packets can be identified by a trial of ${\mathrm{DEC}}_{n_{s}}$. For $i=1,\ldots ,n_{s}$, let $M_{i}$ be the number of non-redundant coded packets in the *i*th batch. We know that $\sum_{i=1}^{n_{s}}M_{i}=K$. Denote by ${\mathrm{DEC}}_{n_{s}}^{*}$ the same decoder as ${\mathrm{DEC}}_{n_{s}}$ except that the redundant coded packets are removed from the decoder input.

Now, we can construct the systematic outer code. For the systematic outer code, the encoding at the source node works as follows:Partition the message packets B into $n_{s}$ subsets ${\tilde{X}}_{i}$, $i=1,\ldots ,n_{s}$, where the number of packets in the *i*th subset ${\tilde{X}}_{i}$ is $M_{i}$;Calculate $\tilde{B}={\mathrm{DEC}}_{n_{s}}^{*}\left({\tilde{X}}_{1},\ldots ,{\tilde{X}}_{n_{s}}\right)={\mathrm{DEC}}_{n_{s}}^{*}\left(B\right)$;Generate the first $n_{s}$ batches ${\mathrm{ENC}}_{n_{s}}\left(\tilde{B}\right)$;Generate more batches by performing ${\mathrm{ENC}}_{n_{s}^{+}}$ on $\tilde{B}$.

See Figure 2b for an illustration of the above encoding process.

We justify that the above encoding process is systematic by showing that the first $n_{s}$ batches include all the message packets. Denote by ${\mathrm{ENC}}_{n_{s}}^{*}$ the encoder that generates only the $M_{i}$ non-redundant coded packets in the *i*th batch, where $i=1,\ldots ,n_{s}$. For any *K* packets B, ${\mathrm{DEC}}_{n_{s}}^{*}\left({\mathrm{ENC}}_{n_{s}}^{*}\left(B\right)\right)={\mathrm{DEC}}_{n_{s}}\left({\mathrm{ENC}}_{n_{s}}\left(B\right)\right)=B$. Note that ${\mathrm{ENC}}_{K}$ and ${\mathrm{DEC}}_{K}$ can be expressed as square matrices that are inverse to each other, and hence their order can be interchanged without changing the output, i.e., ${\mathrm{ENC}}_{n_{s}}^{*}\left({\mathrm{DEC}}_{n_{s}}^{*}\left(B\right)\right)={\mathrm{ENC}}_{n_{s}}^{*}\left(\tilde{B}\right)=B$.

The computation cost of the third step of encoding can be simplified as not all the packets in the systematic batches need to be regenerated. Let $\left(X_{1},\ldots ,X_{n_{s}}\right)={\mathrm{ENC}}_{n_{s}}\left(\tilde{B}\right).$ We have ${\tilde{X}}_{i}\subset X_{i}$ and $X_{i}\setminus {\tilde{X}}_{i}$ includes only the redundant packets for ${\mathrm{DEC}}_{n_{s}}$ in the *i*th batch. As ${\tilde{X}}_{i}$ is a subset of the message packets, it is not necessary to generate it again. Denote by ${\mathrm{ENC}}_{n}^{-}$ the encoder of $\tilde{B}$ that generates only $X_{i}\setminus {\tilde{X}}_{i}$ for i=1,…,n. Let $\left({\bar{X}}_{1},\ldots ,{\bar{X}}_{n}\right)={\mathrm{ENC}}_{n}^{-}\left(\tilde{B}\right).$ Then, the *n* systematic batches are $X_{i}={\tilde{X}}_{i}\cup {\bar{X}}_{i}$.

The batches generated by the above systematic encoding process will be further transmitted through a network and processed by the inner code. Let $Y^{\prime }$ be the coded packets received by the destination node. To decode, first, DEC is applied on $Y^{\prime }$ to output $\tilde{B}$. Then, we apply ${\mathrm{ENC}}_{n_{s}}$ on $\tilde{B}$ to recover B. See Figure 2c for an illustration of the decoding process.

### 3.3. Computation Cost

At first, it seems that the systematic outer code increases the encoding and decoding computation cost because an additional decoding step is employed in the systematic encoding, and an additional ordinary encoding step is employed in the systematic decoding. However, after careful evaluation, we see that the encoding computation cost of the systematic outer code is lower than that of the ordinary outer code. The decoding computation cost of the systematic outer code depends on the number of message packets received at the destination node. In the worst case, where no message packets are received, the decoding computation cost is doubled.

To assist our discussion, we denote by *b* the average computation cost of encoding a packet using the ordinary outer code, and we denote by *c* the computation cost of decoding the ordinary outer code using *K* coded packets. Here, we assume that the decoding is successful with zero coding overhead. Suppose that the packet length *T* is much larger than *M*, which means that the coefficient vector length is much less than *T*. According to the analysis in [23], b=O(M) and c=O(KM) linear combination operations (LCOs). (A linear combination operation (LCO) refers to the computation of a linear combination x+αy, where x and y are two packets of *T* field elements and α is an element from the base field.) Moreover, for the two-step decoding and the joint decoding, Kb≈c. For the inactivation decoding, if the number of inactive packets is bounded by a constant, Kb≈c.

#### 3.3.1. Encoding Computation Cost

The encoding computation cost depends on the number of coded packets generated. For the ordinary outer encoding, the computation cost of encoding *k* packets is kb, where k=1,2,…. For the systematic outer code, we assume $n_{s}M=K$ (we will discuss how to design such a code). As the first *K* packets are the message packets, the encoding of the first *K* packets requires no computation. To encode more packets, the systematic outer code needs to execute ${\mathrm{DEC}}_{n_{s}}^{*}$, which has a computation cost *c*, and ${\mathrm{ENC}}_{n_{s}^{+}}$, which takes computation cost *b* on average to generate a packet. Therefore, when k>K, the computation cost of generating the first *k* coded packets using the systematic outer code is (k−K)b+c≈kb. See the illustration in Figure 3a regarding the computation cost of generating the first *k* packets.

To further understand how the encoding computation cost affects the operation at the source node, we consider two models of message packet arrival at the source node. In the first model, the message packets arrive one-by-one with a unit time interval between two consecutive packets. The ordinary outer code encoding can only start to generate the first coded packets from the time *K* when a precode with HDPC is employed. Let Δ be the time taken by the ordinary encoder to generate *K* coded packets, where Δ∝Kb≈c. The systematic outer code can generate a coded packet upon the arrival of each message packet. At the time *K*, the systematic outer code executes ${\mathrm{DEC}}_{n_{s}}^{*}$, which also takes Δ time. In the second model, all the *K* message packets arrive together at the same time, e.g., time *K*. For this model, the ordinary outer code behaves in the same way as for the previous model, and the systematic outer code can generate the first *K* coded packets at time *K*.

We see that for both message packet arrival models, the systematic outer code generates the first *K* packets earlier than the ordinary outer code. When k>K, both encoders generate the *k*th packet at the same time. See an illustration of this in Figure 3b.

#### 3.3.2. Decoding Computation Cost

For the systematic outer code, the decoding computation cost depends on the number $K_{m}$ of message packets received by the destination node. When $K_{m}=K$, i.e., all the message packets are received, no computation is required for decoding. When $K_{m}<K$, the systematic code decoder needs to execute DEC, which has a computation cost *c*, and ${\mathrm{ENC}}_{n_{s}}^{*}$, which takes computation cost *b* on average to generate a packet. As $K_{m}$ message packets have been received, we only need to use ${\mathrm{ENC}}_{n_{s}}^{*}$ to generate the remaining $K-K_{m}$ message packets. Therefore, the overall decoding computation cost is $\left(K-K_{m}\right)b+c\approx 2c-K_{m}b$. When $K_{m}$ is close to *K*, the systematic outer code decoding computation cost is close to the ordinary outer code decoding. In the worst case, i.e., $K_{m}=0$, the systematic outer code decoding computation cost is doubled compared with the ordinary outer code decoding. See an illustration in Figure 4a.

To illustrate how the decoding computation cost affects the operation at the destination node, we consider that coded packets are received one-by-one with a unit time interval between two consecutive packets. We assume that the ordinary outer code decoder starts decoding at time *K* and takes additional Δ time to decode all the message packets. When $K_{m}=K$, all the message packets are decoded at time *K*. When $K_{m}<K$, the systematic outer code decoder executes DEC at time *K* and starts to use ${\mathrm{ENC}}_{n_{s}}^{*}$ from time K+Δ to generate the $K-K_{m}$ message packets that have not been received. In the worst case, where $K_{m}=0$, the systematic outer code decodes all the message packets at time K+2Δ. See an illustration in Figure 4b.

### 3.4. Random Design

To implement the general approach to the systematic outer code, we only need to design a consistent pair (ENC,DEC). In the following part of this section, we discuss the traditional random approach to designing a consistent (ENC,DEC). In the next section, we discuss a new approach that can design a consistent (ENC,DEC) more effectively.

Denote by h the rank distribution of the batches and let $\Psi^{A}$ be the degree distribution optimized for h as in ([23], Chapter 6), which achieves the near-to-optimal rate of the ordinary outer code as in Section 2.1 asymptotically. We can use the ordinary encoder and decoder as introduced in Section 2.1 with the degree distribution $\Psi^{A}$ to design ${\mathrm{ENC}}_{n_{s}^{+}}$ and DEC for a consistent outer code (ENC,DEC).

The ordinary outer code is random, but we need a deterministic encoder–decoder pair $\left({\mathrm{ENC}}_{n_{s}},{\mathrm{DEC}}_{n_{s}}\right)$ to satisfy the consistent properties. For a given $n_{s}\geq K/E\left[h\right]$, we can perform random trials of the ordinary outer code using the degree distribution $\Psi^{A}$ until an instance $\left({\mathrm{ENC}}_{n_{s}},{\mathrm{DEC}}_{n_{s}}\right)$ is found such that (Equation 3) is satisfied. Note that it is sufficient for us to find only one such instance. As both ${\mathrm{ENC}}_{n_{s}}$ and ${\mathrm{ENC}}_{n_{s}^{+}}$ generate batch instances following the random outer code encoder with the degree distribution $\Psi^{A}$, which is optimized for h, DEC can guarantee a high decoding success probability for a sufficiently large number of received batches [23].

If such an instance cannot be found for a certain value $n_{s}$, we can increase the value of $n_{s}$ by 1 and try again. The ordinary outer code is expected to decode correctly with a high probability when the number of batches is sufficiently large, and we expect to design a systematic code with the *expected coding overhead* $n_{s}E\left[h\right]-K$ as small as possible.

When M=1 and E[h]=1, i.e., the case of fountain codes, a consistent outer code exists for a range of the values of *K* when $n_{s}=K$ using this approach [26]. For fountain codes, the random design works well as fountain codes have a universal design that can handle all packet loss patterns. The random design is only performed once for each value of the number of message packets *K*. Therefore, the efficiency of the random design is not an issue. In other words, a large number of random trials can be performed to find a consistent outer code with a small or zero coding overhead.

Although the random design is suitable for fountain codes, it can be less efficient when M>1. BATS codes are not universal in the sense that the optimal degree distribution depends on the rank distribution h. Therefore, even for the same value of *K*, the random design needs to be repeated for each h, and this may need to be carried out for h obtained online. Hence, the efficiency of the random design becomes an issue for BATS codes with batch size M≥2. For M=16, we perform some experiments using the BATS code implementation in [45] with the parameters in Appendix B. Inactivation decoding is applied to achieve a lower coding overhead. To limit the computation cost of inactivation decoding, the number of inactive packets is limited to 150. In the experiments, we use the rank distribution h with E[h]=M, which is also called the rank-*M* distribution. The experimental results are summarized in Table 1. We observe that when *K* is up to 400M, a consistent instance with $n_{s}=K/M$ can be found. However, the larger the value of *K*, the lower the probability of a code with zero coding overhead. For example, when K=10M, most instances have zero overhead. Meanwhile, when K=400M, only four instances have zero coding overhead. However, when *K* is 600M, no instance is found with zero coding overhead.

## 4. Triangular Embedding: A Structured Systematic Outer Code Design

We propose a structured design of consistent outer codes with a general batch size M≥1. Our approach is based on the following observation. For a consistent instance found by the random design, ${\mathrm{DEC}}_{n_{s}}$ gives an order of the batches such that the *i*th batch is solvable if all the previous batches are solved. Our approach, called *triangular embedding*, tries to design ${\mathrm{ENC}}_{n_{s}}$ so that the order of the batches for solving is predefined. When M=1, our approach also gives a new design of systematic fountain codes.

### 4.1. Triangular Embedding Design

Consider the encoding of *K* message packets with respect to a general rank distribution h. We discuss how to generate the first $n_{s}$ batches, where $n_{s}\geq K/E\left[h\right]$. The precode is the same as the ordinary outer code. Let *K* and $K^{\prime }$ be the number of message packets and the number of precoded packets, respectively. The precoded packets are also separated into active and inactive packets. Let *A* be the number of active packets, where A≥K.

For the degree distribution Ψ optimized for h, we assume that the degree probability is zero for degrees from 1 to M−1 (This assumption does not affect the generality of our design as it is asymptotically optimal to use such a degree distribution when the rank *M* probability of the batch transfer matrix is positive. When the probability of transfer matrix rank *M* is zero, we should reduce the batch size to improve the network’s communication efficiency). Generate the active degree values $d_{1}^{A},\ldots ,d_{n_{s}}^{A}$ for the first $n_{s}$ batches by sampling Ψ. To simplify the discussion, we assume that the degree values are ordered so that $M\leq d_{1}^{A}\leq d_{2}^{A}\leq \cdots \leq d_{n_{s}}^{A}$. The inactive degree $d_{i}^{B}$ of the *i*th batch is obtained in the same way as the ordinary outer code.

Fix positive integers $M_{1},M_{2},\ldots ,M_{n_{s}}$ such that $\sum_{i=1}^{n_{s}}M_{i}=K$ and $M_{i}\leq M\leq d_{i}^{A}$. For example, when $n_{s}$ divides *K*, we may choose $M_{i}=K/n_{s}$. When $n_{s}$ does not divide *K*, there exist unique non-negative integers *a* and $b<n_{s}$ such that $K=an_{s}+b.$ We may let $M_{i}=a+1$ for i=1,…,b and let $M_{i}=a$ for $i=b+1,\ldots ,n_{s}$.

Let $N_{\mathrm{inac}}$ be the maximum number of dynamic inactive packets allowed during inactivation decoding. We should determine the total number of inactive packets $N_{\mathrm{inac}}+K^{\prime }-A$ according to the decoding computation cost constraint. For example, $N_{\mathrm{inac}}+K^{\prime }-A=2\left\lceil \sqrt{K}\right\rceil$. We further assume that for $i=1,\ldots ,n_{s}$,
(4)$$d_{i}^{A}\leq \min\left\{A-K,N_{\mathrm{inac}}\right\}+\sum_{j=1}^{i}M_{j}.$$

This assumption is usually satisfied by the degrees sampled as 1) A−K is linear in *K* and 2) the average degree of a BATS code degree distribution is only around two times the batch size *M* and even the maximum degree is O(M). If $d_{i}^{A}$ does not satisfy (Equation 4), which should occur rarely, we can modify $d_{i}^{A}$ to this upper bound or re-sample the active degrees.

Let $m_{1}=0$, and, for i≥2, let $m_{i}=m_{i-1}+M_{i-1}$. For i≥1, the $d_{i}^{A}$ active packets in $B_{i}$ include

the $\left(m_{i}+1\right)$th to the $\left(m_{i}+M_{i}\right)$th active packets, anda set of $d_{i}^{A}-M_{i}$ packets chosen from the first $m_{i}$ active packets and the last $\min\{A-K,N_{\mathrm{inac}}\}$ active packets.

The inactive packets in $B_{i}$ are obtained in the same way as the ordinary outer code. The *i*th batch is generated as $B_{i}G_{i}$, where $G_{i}$ is a $d_{i}\times M$ matrix different from the ordinary outer code. The rows of $G_{i}$ corresponding to the $\left(m_{i}+1\right)$th to the $\left(m_{i}+M_{i}\right)$th active packets have the form $\left[\begin{array}{cc} I_{M_{i}} & U \end{array}\right]$, where $I_{M_{i}}$ is the $M_{i}\times M_{i}$ identity matrix and U is the $M_{i}\times \left(M-M_{i}\right)$ uniformly random matrix. The other rows of $G_{i}$ are uniformly random. The batches generated can be transmitted following an arbitrary order.

Define ${\tilde{G}}_{i}$ as the $K^{\prime }\times M$ matrix by inserting zero rows into $G_{i}$ so that $\left[B\;B_{p}\right]{\tilde{G}}_{i}=B_{i}G_{i}$. The *overall generator matrix* of ${\mathrm{ENC}}_{n_{s}}$ can be written as $\tilde{G}=\left[\begin{array}{ccc} {\tilde{G}}_{1} & \cdots & {\tilde{G}}_{n_{s}} \end{array}\right]$. According to the design of ${\mathrm{ENC}}_{n_{s}}$, $\tilde{G}$ is of the form in Figure 5.

### 4.2. Decoder Design

It is possible to use the decoders discussed in Section 2.3 to decode the batches generated by triangular embedding. However, due to the structure of the triangular embedding encoding, the decoder can be simplified.

We design a decoder ${\mathrm{DEC}}_{n_{s}}$ using only the first $M_{i}$ packets of the *i*th batch, $i=1,\ldots ,n_{s}$. The overall generator matrix ${\tilde{G}}^{*}$ of ${\mathrm{ENC}}_{n_{s}}^{*}$ is of the form in Figure 6.

An inactivation decoder can be applied to decode the message packets:First, inactivate all the packets used by the first $n_{s}$ batches among the last A−K active packets;Second, apply belief propagation decoding to solve all the batches;Last, solve the inactive packets.

Note that as, at most, $\min\{A-K,N_{\mathrm{inac}}\}$ packets are used among the last A−K active packets during encoding, the total number of inactive packets is no more than $N_{\mathrm{inac}}+K^{\prime }-A$.

### 4.3. Design Verification

We verify the triangular embedding design from two aspects. First, it can help to generate zero-coding-overhead consistent outer codes using a small number of random trials. Second, when jointly decoded with batches generated by the ordinary outer code, the decoding performance is similar to the case of decoding only the batches generated by the ordinary outer code.

We perform the experiments using the batch size M=16 and the rank-*M* rank distribution h. As with the experiments in Section 3.4, we use the BATS code implementation in [45] with the parameters in Appendix B. The experimental results of the triangular embedding outer code are shown in Table 2. We see that, using triangular embedding, for *K* up to 1000M, more than 99.5% instances are of zero coding overhead. In fact, for the remaining instances that are not of zero coding overhead, the coding overhead is only 1 packet (generated using the ordinary outer code). The last row in Table 2 gives the maximum number of inactive packets for all the instances tested for each value of *K*. We see that the number of inactivations is lower than 150, the number of inactivations in the random design. Therefore, diagonal embedding also reduces the decoding computation cost.

As ${\mathrm{ENC}}_{n_{s}}$ uses a different encoding approach to the ordinary outer code ${\mathrm{ENC}}_{n_{s}+}$, we consider whether the batches generated by diagonal embedding and the batches generated by the ordinary outer code together form a good outer code. We perform some numerical experiments to verify the joint decoding performance of these two types of batches. For each batch generated by triangular embedding, we discard the batch with probability ϵ=0.1,0.3,0.5 and send the remaining batches to the decoder. After the first $n_{s}$ batches, the ordinary outer code is applied to generate more batches for the decoder. We adopt the same degree distribution Ψ optimized for the rank-*M* distribution. The results are shown in Table 3. We see that for all the cases of ϵ and for K=10M,100M,200M, the number of zero-coding-overhead instances is higher than that in Table 1 and the number of instances with a coding overhead larger than 2M is lower than that in Table 1. For K=400M,600M, the decoding performance is similar to that in Table 1 in terms of both the ratio of zero coding overhead and the ratio of coding overhead larger than 2M.

## 5. Inner Code for Systematic Batches

In this section, we study the design of the inner code for systematic batches. Based on the discussion in Section 3.3, the decoding complexity at the destination node depends on the number of message packets received. However, using the existing inner coding schemes, the number of message packets in a systematic batch reduces significantly during communication. In the worst case, when no message packets are received at the destination node, the decoding computation cost at the destination node is doubled when compared with the ordinary BATS outer code. To resolve this issue, we discuss how to design the inner code to preserve the message packets in systematic batches.

### 5.1. Detailed Inner Code Formulation

We first formulate in detail how each network node performs the inner code. We also discuss the existing inner coding schemes for systematic batches.

We consider the inner code on a line network as described in Section 2. As the inner code is performed on each batch individually, we consider a generic systematic batch X without the subscripts. We assume that the packets in X are all message packets. By (Equation 2), the received packets $Y^{\left(u\right)}$ of the batch X at node *u* satisfy
(5)$$Y^{\left(u\right)}=XH^{\left(u\right)},$$
where $H^{\left(u\right)}$ is the transfer matrix of the batch at the node *u*.

Let $N_{u}$ be the number of columns of $Y^{\left(u\right)}$ (or $H^{\left(u\right)}$), i.e., the number of received packets of the batch at node *u*. For a non-destination node *u*, we use u+ to denote the receiver of the outgoing link of *u* in the line network. Suppose that the node *u* needs to transmit $N_{u}^{\prime }$ packets of the batch X to the node u+. The transmitted packets, called *recoded packets*, are generated by linear combinations as $Y^{\left(u\right)}\Phi^{\left(u\right)}=XH^{\left(u\right)}\Phi^{\left(u\right)}$, where $\Phi^{\left(u\right)}$ is an $N_{u}\times N_{u}^{\prime }$ matrix over the base field $F_{q}$. Due to packet loss, the set of received packets at u+ is a subset of $Y^{\left(u\right)}\Phi^{\left(u\right)}$. Let $E^{\left(u\right)}$ be an $N_{u}^{\prime }\times N_{u+}$ matrix obtained by removing the columns of an identity matrix specifying the packet erasures. We can write
(6)$$Y^{\left(u+\right)}=XH^{\left(u\right)}\Phi^{\left(u\right)}E^{\left(u\right)}=XH^{\left(u+\right)},$$
where $H^{\left(u+\right)}=H^{\left(u\right)}\Phi^{\left(u\right)}E^{\left(u\right)}$.

There are many solutions to design the recoding matrix $\Phi^{\left(u\right)}$ in the literature. One common method for RLNC is a uniformly random matrix over the base field, which is also called the *random linear inner code* (RLIC). For multicast communications, it has been shown that RLIC achieves the multicast capacity for networks with packet loss [5,8,9,10]. For the line network discussed here, the *systematic inner code* (SIC) has been proposed [23], where all the linearly independent received packets are directly used as recoded packets. We first discuss the performance of these two existing inner code schemes for systematic batches.

When using RLIC for systematic batches, the probability that a recoded packet (a column of $X\Phi^{\left(u\right)}$) is a message packet is $q^{-M}$.When using SIC for systematic batches, if the network links have no packet loss, the destination node receives all the message packets without decoding. If each link has an erasure probability ϵ>0 independently, the number of received message packets at the destination node drops exponentially rapidly with *L* increasing.

In other words, for both RLIC and SIC, the destination node cannot benefit from the systematic outer code.

We are motivated to study the recoding $\Phi^{\left(u\right)}$ such that a non-source node *u* can receive more message packets from a systematic batch even when there are packet losses.

### 5.2. Recovery of Individual Message Packets

Although $Y^{\left(u\right)}$ does not include any message packets, it may be possible to decode some message packets from (Equation 5). When $\mathrm{rank}(H^{\left(u\right)})=M$, all the message packets of a batch can be decoded at node *u*. Note that for batched network coding, $H^{\left(u\right)}$ does not necessarily need to be of rank *M*. We say that a message packet, i.e., a column of X, can be recovered at node *u* if it can be uniquely solved from the system (Equation 5). When $\mathrm{rank}(H^{\left(u\right)})<M$, some of the message packets can be recovered by operations within a systematic batch.

Denote by $\mathrm{Col}(H^{\left(u\right)})$ the column space of the matrix $H^{\left(u\right)}$. Let $e_{i}$ be the length-*M* column vector with its *i*th entry 1 and all the other entries 0. A necessary and sufficient condition such that a message packet can be recovered from $Y^{\left(u\right)}$ is as follows.

**Lemma 1.** 
*Under the condition that $Y^{\left(u\right)}=XH^{\left(u\right)}$ is consistent, the ith packet in X has a unique solution if and only if $e_{i}\in \mathrm{Col}\left(H^{\left(u\right)}\right)$.*


**Proof.** The lemma can be proven by the equivalence of the following statements:
The *i*th packet in X has a unique solution;All the vectors $x\in F_{q}^{N_{u}}$ such that $xH^{\left(u\right)}=0$ (called the left nullspace collectively) have the *i*th entry 0;$e_{i}$ is orthogonal to the left nullspace of $H^{\left(u\right)}$;$e_{i}$ is in the column space of $H^{\left(u\right)}$.   □

The following proposition shows that we can test the recoverability of all the message packets in a systematic batch from the reduced column echelon form of $H^{\left(u\right)}$, which can be obtained by (column-wise) Gauss–Jordan elimination.

**Proposition 1.** 
*Let L be the reduced column echelon form for a matrix $H^{\left(u\right)}$. Then, $e_{i}\in \mathrm{Col}\left(H^{\left(u\right)}\right)$ if and only if $e_{i}$ is a column of L.*


**Proof.** See Appendix A.    □

The next proposition shows that if a message packet cannot be recovered at a node, it cannot be recovered at any of the following nodes. Equivalently, if a message packet can be recovered at a node, it can be recovered at all the previous nodes.

**Proposition 2.** 
*If a message packet cannot be recovered at the node u, then it cannot be recovered at the node u+ on the next hop.*


**Proof.** If the *i*th message packet cannot be recovered at the node *u*, by Lemma 1, we have $e_{i}\notin \mathrm{Col}\left(H^{\left(u\right)}\right)$. Due to $\mathrm{Col}\left(H^{\left(u+\right)}\right)=\mathrm{Col}\left(H^{\left(u\right)}\Phi^{\left(u\right)}E^{\left(u\right)}\right)\subseteq \mathrm{Col}\left(H^{\left(u\right)}\right)$, $e_{i}\notin \mathrm{Col}\left(H^{\left(u+\right)}\right)$ and hence the *i*th message packet cannot be recovered at the node $u^{+}.$    □

In general, performing an elementary operation as used in Gauss–Jordan elimination on the received packets of a batch does not affect the rank of the batch, and hence does not affect the decoding performance. However, recovering message packets at the intermediate nodes helps to improve the number of message packets to be received/recovered in the next hop. We use an example to illustrate this fact.

**Example 1.** 
*Consider a line network with L=2, M=3 and $N_{1}=M$ at node 1. Suppose that*

$$H^{\left(1\right)}=\left[\begin{array}{ccc} 1 & 0 & a\\ 0 & 1 & b\\ 0 & 0 & c \end{array}\right],$$

*where a,b,c≠0 are elements from the base field. Using systematic recoding on $H^{\left(1\right)}$, no additional packets should be generated and $Y^{\left(1\right)}$ is transmitted by node 1. When the second packet is lost from node 1 to 2, we obtain*

$$H^{\left(2\right)}=\left[\begin{array}{cc} 1 & a\\ 0 & b\\ 0 & c \end{array}\right],$$

*At destination node 2, we can only recover one message packet. On the other hand, suppose that we apply the Gaussian elimination step at node *1* and the result should be $H^{\left(1\right)}D=I$. Then, node *1* transmits $Y^{\left(1\right)}D$ instead of $Y^{\left(1\right)}$. In this case, if we still erase the second packet, the following node can recover *2* message packets. Moreover, since the Gaussian elimination step preserves the column space of the batch transfer matrix, ($\mathrm{Col}\left(H^{\left(u\right)}\right)=\mathrm{Col}\left(H^{\left(u\right)}D\right)$), the rank and number of recoverable message packets at the destination node should be at least as good as in the recoding schemes without this step.*


Note that the recovery of the message packets at an intermediate node is a linear operation on a batch and hence can be regarded as a part of the inner code. The effect of the recovery of the message packets can be captured by the coefficient vectors: the same operation applied on the received packets of a batch is applied on the coefficient vectors as well. The destination node does not need to know the exact operations at each intermediate, but only the coefficient vectors of the received packets.

### 5.3. Side Information for Message Packet Recovery

We discuss some general properties involved in the recovery of message packets at the node u+, which provide guidance for the design of new inner codes. The recoverability of a message packet depends on the knowledge of $H^{\left(u\right)}$, which is delivered by the coefficient vectors. Note that the original purpose of the coefficient vectors is for the destination node to decode the batches. A natural question to consider is the following: if more information is delivered from node *u* to u+, could more message packets be recovered at node u+?

**Proposition 3.** 
*Suppose that X, $H^{\left(u\right)}$, $\Phi^{\left(u\right)}$ and $E^{\left(u\right)}$ in (Equation 6) are mutually independent. $\Phi^{\left(u\right)}$ and X are conditionally independent given $H^{\left(u+\right)}$ and $Y^{\left(u+\right)}$.*


**Proof.** See Appendix A.    □

The above proposition states that $\Phi^{\left(u\right)}$ and X are conditionally independent at the node u+. The next proposition further shows that knowing $\Phi^{\left(u\right)}$ at the node u+ does not help to recover more message packets at node u+. It actually shows a stronger result that knowing any variable that is independent with X given $H^{\left(u+\right)}$ and $Y^{\left(u+\right)}$ at the node u+ does not help in recovering more message packets at the node u+.

**Proposition 4.** 
*Suppose that X, $H^{\left(u\right)}$, $\Phi^{\left(u\right)}$ and $E^{\left(u\right)}$ in (Equation 6) are mutually independent. Let S be any random variable that is conditionally independent with X given $H^{\left(u+\right)}$ and $Y^{\left(u+\right)}$. Given the instance of $H^{\left(u+\right)}$ and $Y^{\left(u+\right)}$ at the node u+, further knowing the instance of S at the node u+ does not help to recover more message packets at u+.*


**Proof.** See Appendix A.    □

Based on the above analysis, we know that the existing coefficient vectors are sufficient for the recovery of message packets at the intermediate nodes. In other words, it is not necessary for a network node to add further information to assist the recovery of the message packets in the following nodes.

### 5.4. Recoding with Message Packet Protection

Let $r=\mathrm{rank}(H^{\left(u\right)})$ and V be an $N_{u}\times N_{u}$ matrix such that $H^{\left(u\right)}V$ is in reduced column echelon form. To recover message packets, we perform the same column operations on $Y^{\left(u\right)}$ and obtain $Y^{\left(u\right)}V=XH^{\left(u\right)}V$. If $e_{i}$ is the *j*th column of $H^{\left(u\right)}V$, then the *j*th column of $Y^{\left(u\right)}V$ is equal to the *i*th message packet.

Let *s* be the number of message packets that can be recovered from $Y^{\left(u\right)}$. By Proposition 1, there are exactly *s* distinct columns in $H^{\left(u\right)}V$ with only 1 non-zero entry being one. Therefore, by proper row and column permutations, $H^{\left(u\right)}V$ is of the form
(7)$$\left[\begin{array}{ccc} I_{s} & 0 & 0\\ 0 & I_{r-s} & 0\\ 0 & T & 0 \end{array}\right],$$
where $I_{k}$ is the k×k identity matrix, 0 is an all-zero matrix of proper size, and T is an $(N_{u}-r)\times r$ matrix where each column is not zero.

Denoting the first *r* columns of the corresponding column permutation matrix as the $N_{u}\times r$ matrix P, each of the first *s* columns of $H^{\left(u\right)}VP$ has only 1 non-zero entry.

We have discussed the decoding step, which is represented by V. However, to generate a recoded batch, some redundant packets are to be generated. The following proposition states that using the random linear inner code at node *u*, the node u+ can recover almost no message packets when the number of received packets at u+ is fewer than $\mathrm{rank}(H^{\left(u\right)})$. Denote by $\zeta_{k}^{m,n}$ the probability that an m×n uniformly random matrix over $F_{q}$ has rank *k*. See, e.g., ([23], Section 3.3.2) for the formula of $\zeta_{k}^{m,n}$.

**Proposition 5.** 
*Suppose that the random linear inner code over $F_{q}$ is used at the node u and $N_{u+}<r=\mathrm{rank}\left(H^{\left(u\right)}\right)$. Under the condition that $e_{i}\in \mathrm{Col}\left(H^{\left(u\right)}\right)$, the probability that $e_{i}\in \mathrm{Col}\left(H^{\left(u+\right)}\right)$ is $1-\sum_{k=0}^{N_{u+}}\zeta_{k}^{r-1,N_{u+}}q^{k-N_{u+}}$ and it converges to zero as q→∞.*


**Proof.** See Appendix A.    □

It is unavoidable that the number of received packets at u+ is smaller than $\mathrm{rank}(H^{\left(u\right)})$ due to packet loss. Together with Proposition 2, Proposition 5 implies that as long as the event $N_{u+}<\mathrm{rank}\left(H^{\left(u\right)}\right)$ occurs once at some node *u*, the destination node receives almost no message packets from a systematic batch. Therefore, random linear recoding is not preferred for the recovery of message packets. Thus, we are motivated to extend systematic inner codes for the recovery of message packets.

We propose two designs of recoding that can protect the message packets during recoding. We first define two recoding matrices. Suppose that *s* message packets are recoverable at the node *u* and the rank of the batch is *r*.


**Message Protection Recoding**


For an integer *w* with $0\leq w\leq N_{u}^{\prime }-s$, let R be an $r\times N_{u}^{\prime }$ matrix of the form
$$R=\left[\begin{array}{ccc} I_{s} & U_{s,w} & U_{r,N_{u}^{\prime }-s-w}\\ 0 & 0 \end{array}\right],$$
where $U_{m,n}$ is the m×n uniformly random matrix.


**Systematic Message Protection Recoding**


For an integer *w* with $0\leq w\leq N_{u}^{\prime }-s$, let $R^{\mathrm{sys}}$ be an $r\times N_{u}^{\prime }$ matrix of the form: when $w<N_{u}^{\prime }-r$,
$$R^{\mathrm{sys}}=\left[\begin{array}{ccc} I_{s} & U_{s,w} & 0\\ 0 & 0 & I_{r-s} \end{array}U_{r,N_{u}^{\prime }-r-w}\right];$$
when $w\geq N_{u}^{\prime }-r$,
$$R^{\mathrm{sys}}=\left[\begin{array}{ccc} I_{s} & U_{s,w} & 0\\ 0 & 0 & J \end{array}\right],$$
where J is the first $N_{u}^{\prime }-w-s$ columns of $I_{r-s}$.

The inner code operations at node *u* consist of (i) the Gauss–Jordan elimination represented by the matrix V, (ii) the column permutation and removal of the all-zero columns represented by the matrix P, and (iii) (systematic) message protection recoding R ($R^{\mathrm{sys}}$). When the overall recoding matrix at node *u* is VPR, the inner code is called the *message protection inner code (MPIC)*. When the overall recoding matrix at node *u* is $VPR^{\mathrm{sys}}$, the inner code is called the *systematic message protection inner code (SMPIC)*.

The value of *w* controls the level of protection of message packets. When w=0, no additional protection is provided for message packets, and we can check that SMPIC has the same rank performance as the systematic inner code. When $w=N_{u}^{\prime }-s$, all recoded packets generated by linear combinations are used to protect the message packets.

The computation cost of the proposed message protection recoding at a network node is mainly determined by (1) the Gauss–Jordan elimination for the recovery of the message packets, and (2) the generation of the recoded packets. To simplify the discussion, we only consider the case with w=0. At node *u*, Gauss–Jordan elimination is applied on the $N_{u}$ received packets. As the packet length *T* is much larger than the batch size *M*, the computation cost of processing the transfer matrix $H^{\left(u\right)}$ is ignored. Hence, when the rank of $H^{\left(u\right)}$ is *r*, the computation cost of recovering the message packets is about $r(N_{u}-1)$ LCOs. If the previous node also uses message protection recoding, the cost at node *u* can be reduced, as the message packets received directly can help to simplify the Gauss–Jordan elimination. Let $s_{0}$ be the message packet received at node *u*, and we have $s_{0}\leq s\leq r$. In this case, the computation cost for Gauss–Jordan elimination is about $\left(r-s_{0}\right)\left(N_{u}-1\right)$ LCOs. For a batch of rank *r*, the cost of generating recoded packets using R or $R^{\mathrm{sys}}$ is linear with the number of entries in uniformly random sub-matrices. Therefore, the overall recoding computation cost for SMPIC with w=0 is about $\left(\left(r-s_{0}\right)\left(N_{u}-1\right)+r\left(N_{u}^{\prime }-r\right)\right)$ LCOs. In contrast, for RLIC, the computation cost is $N_{u}^{\prime }N_{u}$ LCOs, and for SIC, the computation cost is $\left(N_{u}^{\prime }-N_{u}\right)N_{u}$ LCOs.

### 5.5. Numerical Evaluations

We perform numerical evaluations to verify the performance of the new inner codes in terms of both the average rank and the average number of recoverable message packets and compare it with that of the random linear inner code (RLIC) and the systematic inner code (SIC). We use line networks of length up to 50 hops, where each link has the same independent packet erasure probability 0.2. The batch size M=16 and the number of packets to transmit $N_{u}^{\prime }=20$ for all nodes *u*. Since the performance of SMPIC and MPIC shows negligible differences in simulation, we only show the results for SMPIC, where we evaluate w=0 and $w=N_{u}^{\prime }-s$ as representatives.

Our numerical evaluation results are shown in Figure 7. We plot the average number of recoverable message packets and the average rank at node 0 to 50 for SIC, RLIC, SMPIC with w=0 (denoted by SMPIC0) and SMPIC with $w=N_{u}^{\prime }-s$ (denoted by SMPIC1), each with 500 trials. We have the following observations.

SIC and RLIC have almost the same rank performance. SIC has a larger number of recoverable message packets than RLIC. However, for both SIC and RLIC, the number of recoverable message packets drops quickly.SMPIC0 has similar rank performance to SIC and RLIC and has a much higher average number of recoverable message packets than that of SIC and RLIC.SMPIC1 has the highest average number of recoverable message packets among the four inner codes, at the cost of a reduced average rank.

The recoding computation costs at each network node are also determined in the experiments and are illustrated in Figure 8. For RLIC, as $N_{u}^{\prime }=20$ and the expectation of $N_{u}=16$, the recoding computation cost is about 320 LCOs. For SIC, the recoding computation cost is about 64 LCOs. The recoding computation cost of SMPIC0 also matches the formula that we have derived, where the expectation of $s_{0}$ is 1−ϵ=0.8 multiplied by the number of recovered message packets in the previous hop. In Figure 8, we also show the computation cost of SMPIC1, which is close to that of SMPIC0.

## 6. Concluding Remarks

In this paper, we propose a design for systematic batched network codes, where the outer code is systematic and the inner code can protect the systematic property during network coding. Our design of the systematic code preserves the most salient features of the BATS code. The diagonal embedding approach is proposed to improve the design efficiency of the systematic outer code, and it can also be used for non-systematic outer coding to reduce the coding overhead and computation cost.

The discussion in this paper can help to evaluate when and how to adopt systematic batched network codes. When the computation cost and the encoding latency are the major concerns, the use of systematic outer codes is preferred due to the lower computation cost compared to the ordinary outer code. The decision regarding whether to use message protection recoding depends on both the computation constraints and the application scenario. When the decoding computation is sensitive and the intermediate nodes have an additional computation capability, it is beneficial to use message protection recoding. Message protection recoding is also preferred for some application scenarios, e.g., for communications where part of the content can be consumed when ready, a systematic code is better. Another useful scenario for systematic codes is a network with dynamic network link qualities: the communication is reliable most of the time and serious packet loss occurs only in a small fraction of the time.

There are still many refinements to be applied for the systematic batched network codes. This paper focused on the inner code design for unicast communications. The current inner codes designed to protect the message packets may not be suitable to achieve the multicast gain of network coding. Further study of the inner code design for multicast communication is desired.

## 7. Patents

Patents resulting from this work are listed in the following:**CN115811381A** The design framework of the systematic BATS code (including the outer code and inner code), invented by the authors of this paper, published on 17 March 2023.**CN2023105394085** The design of the triangular embedded outer code, invented by L.M. and S.Y., filed on 15 May 2023.

## Figures and Tables

**Figure 1 entropy-25-01055-f001:**
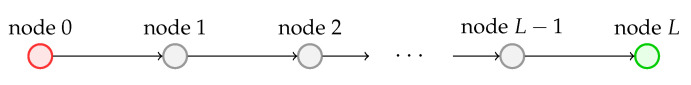
A line network of length *L*. Node 0 is the source node, and node *L* is the destination node. The direct edge from node *i* to node i+1 (i=0,1,…,L−1) illustrates the network link.

**Figure 2 entropy-25-01055-f002:**
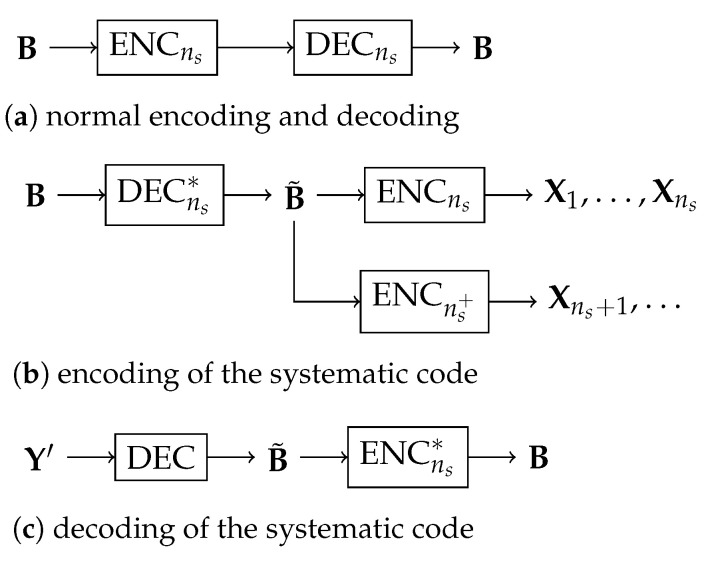
Illustration of the approach to systematic outer codes. (**a**) shows the normal use of a consistent pair of the outer code encoder ${\mathrm{ENC}}_{n_{s}}$ and decoder ${\mathrm{DEC}}_{n_{s}}$. (**b**) shows the encoding of the systematic code, where ${\mathrm{ENC}}_{n_{s}^{+}}$ is the outer code encoder that generates the coded packets beyond the first $n_{s}$ batches. (**c**) shows the decoding of the systematic code, where $Y^{\prime }$ is the received coded packets generated by inner coding.

**Figure 3 entropy-25-01055-f003:**
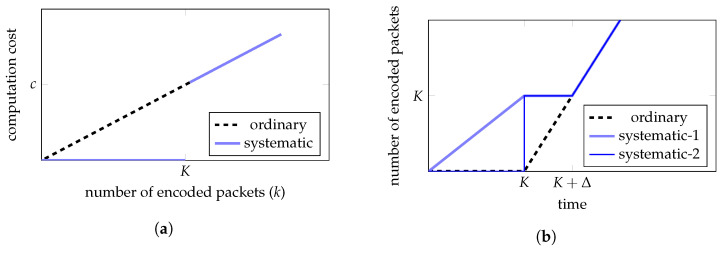
Illustration of the encoding computation cost for the ordinary outer code and the systematic outer code. (**a**) shows the encoding computation cost of generating the first *k* coded packets. For the ordinary outer code, the computation cost increases linearly with *k*. For the systematic outer code, the computation cost is 0 when k≤K. The jump in the computation cost after time *K* is used to execute ${\mathrm{DEC}}_{n_{s}}^{*}$. (**b**) illustrates the number of encoded packets generated over time. The curve “systematic-1” is for the systematic outer code encoder when the message packets arrive one-by-one in each unit time. The curve “systematic-2” is for the systematic outer code encoder when the message packets arrive all at time *K*. From time *K*, these two curves overlap. The ordinary outer code behaves in the same way for both message packet arrival models.

**Figure 4 entropy-25-01055-f004:**
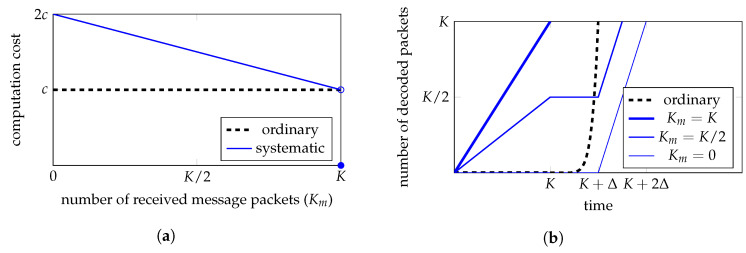
Illustration of the decoding computation costs of the ordinary outer code and the systematic outer code. (**a**) illustrates the decoding computation cost for different numbers $K_{m}$ of message packets received. For the ordinary outer code, the decoding computation cost is *c*. For the systematic outer code, the decoding computation cost is approximately $2c-K_{m}b$ when $K_{m}<K$ and 0 when $K_{m}=K$. (**b**) shows the number of decoded packets over time. The three curves labeled $K_{m}=K,K/2,0$ are for the systematic outer code decoder with $K_{m}$ message packets received.

**Figure 5 entropy-25-01055-f005:**
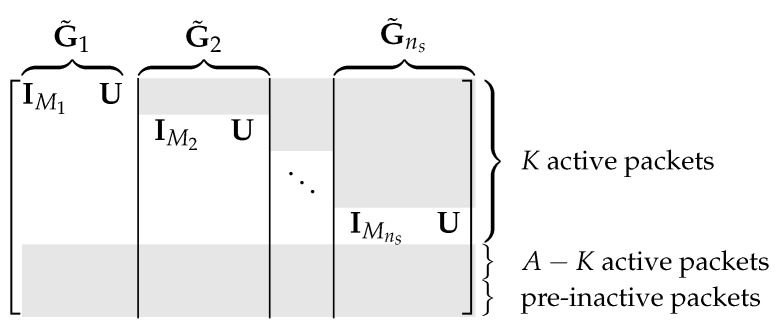
Illustration of encoding using triangular embedding. The gray part contains non-zero entries and the white part contains only zero.

**Figure 6 entropy-25-01055-f006:**
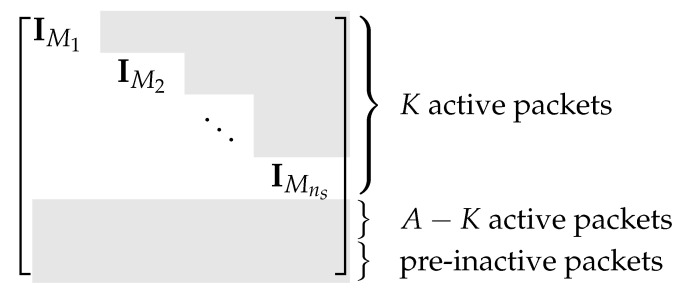
Illustration of decoding using triangular embedding. The gray part contains non-zero entries and the white part contains only zero.

**Figure 7 entropy-25-01055-f007:**
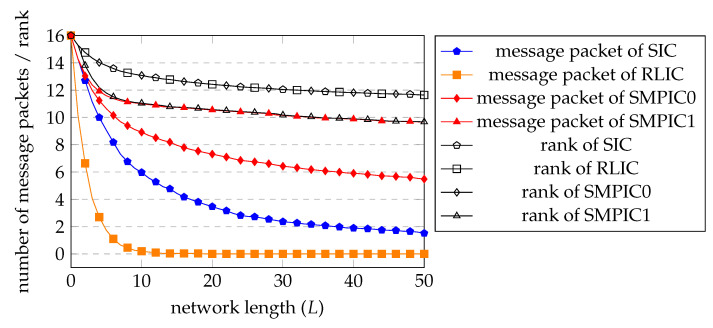
The average number of recovered message packets and the average rank at node 0 to 50 for SIC, RLIC, SMPIC with w=0 (denoted by SMPIC0) and SMPIC with $w=N_{u}^{\prime }-s$ (denoted by SMPIC1), each with 500 trials.

**Figure 8 entropy-25-01055-f008:**
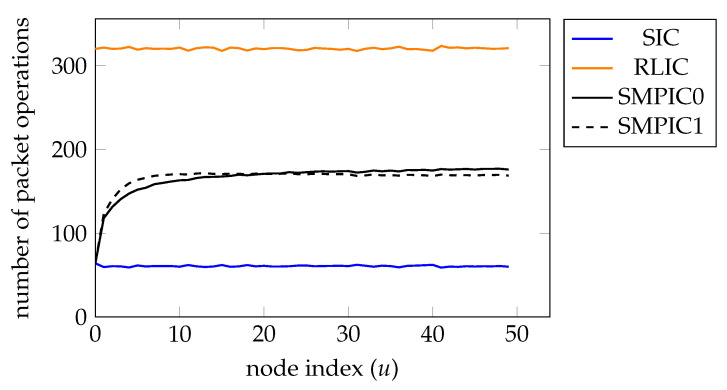
The average number of linear combination operations at node 0 to node 49 for SIC, RLIC, SMPIC with w=0 (denoted by SMPIC0) and SMPIC with $w=N_{u}^{\prime }-s$ (denoted by SMPIC1), each with 500 trials.

**Table 1 entropy-25-01055-t001:** Experiments of random design. Here, M=16 and h has rank *M*. For each value of K=10M,100M,1000M, 5000 instances of the ordinary outer code are sampled. As a BATS code is rateless, for each instance, we can try a range of values of $n_{s}$. The table gives the number of consistent instances when $n_{s}=K/M,K/M+1,K/M+2$, and $n_{s}\geq K/M+3$.

$n_{s}-K/M$	K=10M	K=100M	K=200M	K=400M	K=600M
0	4784	3552	836	4	0
1	173	175	50	0	0
2	34	113	53	0	0
≥3	9	1160	4061	4996	5000

**Table 2 entropy-25-01055-t002:** Experiments using triangular embedding for consistent outer codes. Here, M=16 and h has rank *M* with probability 1. For each value of K=10M,100M,1000M, in total, 5000 instances of the triangular embedded outer code are tested. The table gives the number of consistent instances.

	K=10M	K=100M	K=1000M
zero overhead	4978	4983	4977
max total inact.	27	91	149

**Table 3 entropy-25-01055-t003:** Joint decoding of batches generated by triangular embedding and the ordinary outer code. Here, M=16 and h has rank *M* with probability 1. In our experiments, each batch generated by triangular embedding has a probability ϵ of being discarded, and the remaining batches are sent to the decoder. Following the batches generated by triangular embedding, batches generated by the ordinary outer code are also sent to the decoder. For each value of K=10M,100M,200M,400M,600M and ϵ=0.1,0.3,0.5, in total, 5000 instances are tested.

Coding Overhead	K=10M	K=100M	K=200M	K=400M	K=600M
(**a**) ϵ=0.1					
0	4982	4955	3446	3	0
1∼M	18	26	69	0	0
M+1∼2M	0	3	51	0	0
>2M	0	16	1434	4997	5000
(**b**) ϵ=0.3					
0	4983	4511	1037	17	0
1∼M	17	73	31	0	0
M+1∼2M	0	39	25	1	0
>2M	0	377	3907	4982	5000
(**c**) ϵ=0.5					
0	4987	4155	1823	5	0
1∼M	13	107	101	1	0
M+1∼2M	0	61	72	1	0
>2M	0	677	3004	4993	5000

## Data Availability

Not applicable.

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
