# Peer review of "Design and Analysis of Systematic Batched Network Codes"

_entropy, 2023, doi:10.3390/e25071055_

Round 1

Reviewer 1 Report

This paper introduces the new design to outer and inner code for BATS code to improve the number of message packets received at the receiver. The proposed code construction is reasonable and effective. The presentation of this paper is also of high quality. My main concern is on the complexity of the proposed coding. Please find my detailed comments below:

1. For the proposed outer coding, the decoding is performed before encoding. Based on my understanding, this is to preserve the degree distribution of the BATS code. However, this may introduce substantial complexity. Besides, the intermediate node also needs to perform decoding to increase the number of message packets. This incurs large complexity at the intermediate nodes. The complexity should be explicitly analyzed and compared with the traditional BATS code.

2. The motivation for maintain a large number of message packets should be further highlighted. Since the message packets are not received in order, this may not improve the latency performance. Besides, it does not reduce the decoding complexity either since the complexity is greatly increased to achieve such goal. Without considering the practical benefit of systematic code, it does not make sense to me to use the number message packets as the performance metric.

3. The decoding at the intermediate node may change the coding coefficients of the received packets observed at the receiver. To capture the coding operations at the intermediate node, the transmitter may need to add the overhead at each packet, which can actually be obviated in the traditional BATS code by using synchronized random number generator.

Author Response

Response 1:

We add the complexity analysis and experiments in the revised paper. In Section 3.3, the computation cost of the systematic outer code is analyzed. For encoding, the systematic outer code has a lower computation cost than the corresponding BATS outer code. The decoding computation cost of the systematic outer code depends on the number of message packets received at the destination node. When all the message packets are received, no computation is required for decoding. When some of the message packets are not received, the decoding computation cost of the systematic outer code increases with the number of message packets that are not received and is at most 2 times the computation cost of the BATS outer code decoding.

In Section 5.4, a paragraph is added to analyze the computation cost of the message protection recoding, with a comparison to the existing recoding schemes. In Section 5.5, we further add the computation cost comparison using numerical experiments.

Response 2:

The number of message packets received at the destination node significantly affects the decoding computation cost of the systematic outer code. When some of the message packets are not received, the decoding computation cost of the systematic outer code increases with the number of message packets that are not received.  
However, using the existing inner coding schemes, the number of message packets in a systematic batch reduces significantly during communication. In the worst case, when no message packets are received at the destination node, the decoding computation cost at the destination node is doubled when compared with the ordinary BATS outer code. To resolve this issue, we discuss how to design the inner code to preserve the message packets in systematic batches. 

The above discussed is also added at the beginning of Section 5.

Response 3:

The decoding at the intermediate node (for the purpose of recovery message packets) is also a linear operation and hence can be regarded as a part of the inner code. The coefficient vectors can capture the effect of the recovery of the message packets: The same operation applied on the received packets of a batch is applied to the coefficient vectors as well. For decoding at the destination node, it is not necessary to know the exact operations at each intermediate node, but only the coefficient vectors of the received packets. We add a remark at the end of Section 5.2.

Moreover, as we have analyzed in Section 5.3, the existing coefficient vectors are sufficient for the recovery of message packets at the intermediate nodes. It is not necessary for a network node to add extra information to assist in the recovery of the message packets in the following nodes.

Reviewer 2 Report

The authors study systematic batched network codes that have a systematic outer code and an inner code. Network coding was a hot topic 10 years ago but it is good to see that still some activates are happening and new results are published. The paper is well-written and easy to follow it. I like the given examples.

One suggestion is to write the encoding and decoding algorithms in an algorithmic format.

Another suggestion is to add a paragraph about overhead reduction. There are related works that consider header compression such as

 D. Gligoroski, et al., "Minimal header overhead for random linear network coding," 2015 IEEE International Conference on Communication Workshop (ICCW), London, UK, 2015, pp. 680-685, doi: 10.1109/ICCW.2015.7247260.

Author Response

Response 1:

We add the algorithms of the BATS outer code encoder (ENC) and decoder (DEC) in the algorithmic format in Appendix C. The systematic outer code is constructed using the encoder and decoder of the BATS outer code as illustrated in Figure 2 and explained in Section 3.2: Lines 390—394 for encoding and Lines 409—412 for decoding.

Response 2:

In the first paragraph, we add two sentences about overhead reduction, where some related works, including the one in the comment, are cited.  

Reviewer 3 Report

This work proposes a general systematic outer code construction that achieves minimal computational cost in with respect to encoding/decoding. A triangular embedding approach is also proposed for constructing the systematic batches to reduce the number of random trials for searching a code with a close-to-optimal coding overhead. Some minor comments are as follows:

1. Please state the drawbacks of the proposed approach - i.e., the trade-off for lowering the computational cost.

2. Please  provide all the variable and parameter settings for the numerical experiments so it can be reproduced by other researchers.

3. Please include a nomenclature and abbreviations section - so the manuscript is easier to follow.

4. In the comparative analysis, an introduction of a metric to gauge the optimization would be useful in the current work (or future work). 

Author Response

Response 1:

We compare the computation complexity of the proposed systematic batched code with the existing schemes in the revised paper. In Section 3.3, the computation cost of the systematic outer code is analyzed. For encoding, the systematic outer code has a lower computation cost than the corresponding BATS outer code. The decoding computation cost of the systematic outer code depends on the number of message packets received at the destination node. When all the message packets are received, no computation is required for decoding. When some of the message packets are not received, the decoding computation cost of the systematic outer code increases with the number of message packets that are not received and is at most two times the computation cost of the BATS outer code decoding.

In Section 5.4, a paragraph is added to analyze the computation cost of the message protection recoding, with a comparison to the existing recoding schemes. In Section 5.5, we further add the computation cost comparison using numerical experiments.

Response 2:

For the experiments in Sections 3.4 and 4.3, we provide the code parameters, such as the degree distribution, LDPC, and HDPC numbers, in Appendix B. 

Response 3:

Abbreviations are in the part just before Appendix A.  

Response 4:

We add elaborations about the performance metrics in this revision. In Section 2.3, the outer coding rate and the outer coding overhead are further elaborated. In Section 3.3, about the computation cost comparison, we define the linear combination operation as the unit of the computation cost for the outer code and inner code.

Round 2

Reviewer 1 Report

The authors have added sufficient discussions on the complexity of the proposed design. I am satisfied with the current presentation of the paper.